# Building a stem cell-based primate uterus

Sophie Bergmann [1,2,3], Magdalena Schindler[1,2,3], Clara Munger[1,2,3],
Christopher A. Penfold [1,2,3,4 ✉] & Thorsten E. Boroviak [1,2,3 ✉]

The uterus is the organ for embryo implantation and fetal development. Most current models of the uterus are centred around capturing its function during later stages of pregnancy to increase the survival in pre-term births. However, in vitro models focusing on the uterine tissue itself would allow modelling of pathologies including endometriosis and uterine cancers, and open new avenues to investigate embryo implantation and human development. Motivated by these key questions, we discuss how stem cell-based uteri may be engineered from constituent cell parts, either as advanced self-organising cultures, or by controlled assembly through microfluidic and print-based technologies.

T he primary function of the uterus is to provide a suitable environment for the embryo to implant and gestate to full term. Recent single-cell transcriptional atlases of the uterus during the menstrual cycle[1,2] and of the maternal–foetal interface during the first trimester of pregnancy in human[3,4], provide comprehensive roadmaps to drive forward the development of stem cell-based models. The goal of such stem cell-based uterus models is to engineer a defined, flexible, and scalable system to address fundamental questions of reproductive biology. Major topics include mechanisms and pathologies of embryo implantation, embryogenesis, crosstalk between the developing embryo and the mother, and pathologies of the female reproductive tract. The first trimester of pregnancy is a particularly dynamic and critical stage when the embryo organises the body plan before the foetal growth phase in the second and third trimesters. Although complex in vitro models capable of complete recapitulation of the structure and function of the uterus may be necessary for a clinical setting to increase survival chances of premature fetuses[5], such models would lack the scalability and experimental flexibility required for drug discovery and genetic screening.

In this review, we discuss the construction of stem cell-based uterus models to illuminate the earliest stages of human embryogenesis and associated pathologies. We outline the developmental origin of the uterus and consider its anatomy, function, and diseases. Building on recent breakthroughs in organoid culture, we highlight the constituent components from which a stem cell-based uterus may be engineered and propose a modular approach to assembling uterine models. We discuss approaches based on self-organisation as well as controlled assembly, either through microfluidic or print-based methodologies and conclude how these technologies can be used to tackle questions about implantation failure and key pathologies of the uterus.

## Developmental origin of the uterus

In mammals, the uterus originates from the intermediate mesoderm which is positioned between the paraxial and lateral plate mesoderm after gastrulation[6] (Fig. 1, week 3). At Carnegie stage (CS)10, the embryo folds to form the intraembryonic coeloms which are lined by lateral plate mesoderm and intermediate mesoderm, wherein the inner lining of the coeloms is the coelomic epithelium[7] (Fig. 1, week 5). A subset of intermediate mesoderm cells undergoes mesenchymal-to-epithelial transition to form the nephric duct along the body. This transition requires Pax2

[1] Department of Physiology, Development and Neuroscience, University of Cambridge, Cambridge, UK. [2] Centre for Trophoblast Research, University of Cambridge, Cambridge, UK. [3] Wellcome Trust – Medical Research Council Stem Cell Institute, University of Cambridge, Jeffrey Cheah Biomedical Centre, Cambridge, UK. [4] Wellcome Trust – Cancer Research UK Gurdon Institute, Henry Wellcome Building of Cancer and Developmental Biology, University of Cambridge, Cambridge, UK. ✉email: cap76@cam.ac.uk; teb45@cam.ac.uk

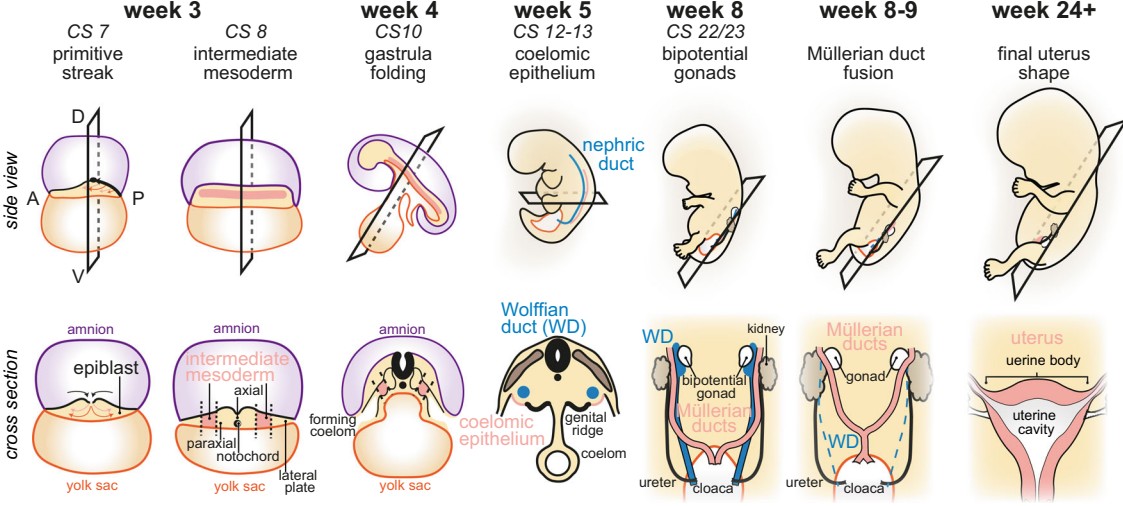

**Fig. 1 Developmental origin of the uterus.** The development of the human uterus is shown in situ (top panel) and cross-sections (bottom panel). Gastrulation and primitive streak formation occur in Carnegie stage (CS)7 embryonic disc 3 weeks post-fertilisation, leading to the formation of intermediate mesoderm at CS 8. At CS10, the embryo has folded and established the intraembryonic coeloms. The nephric ducts develop from the mesodermal coelomic epithelium between CS12 and CS13. The central parts constitute the Wolffian ducts (WD). Concomitantly, the Müllerian ducts invaginate from the coelomic epithelium adjacent to the nephric duct and elongate towards the cloaca. By CS22–23, the bipotential gonads are fully established in proximity to the kidneys. Both are connected to the WDs and Müllerian ducts, ultimately leading to the cloaca. In female fetuses of around week 8–9, the WDs degenerate, and the Müllerian ducts fuse to form the uterine body and the upper vaginal tract. The uterus acquires its final shape at around 24 weeks post-fertilisation.

and Pax8, which induce expression of Lhx1 (Lim1)[8], an important transcription factor of the urogenital system in both mouse and human[9,10]. The nephric ducts are required for the development of adult kidneys, the ureter, and the genital tract. The initially central portion of the nephric duct is known as the Wolffian (mesonephric) ducts.

Morphogenetic rearrangements at CS12–16 lead to the insertion of the Wolffian ducts into the cloaca, the precursor of the bladder[11,12]. Intermediate mesoderm-derived coelomic epithelial cells invaginate to form the Müllerian (paramesonephric) ducts at CS14–17, which elongate caudally along the Wolffian ducts at CS18–23. In mouse, the Müllerian and Wolffian ducts express Lhx1 and Wnt signalling is required for Wolffian duct elongation[13,14]. Histological studies of human embryos suggest that this morphogenetic process is conserved[11,15]. At CS23, Wolffian and Müllerian ducts together form the bipotential genital tract[16]. In males, the Müllerian ducts degenerate and the Wolffian ducts persist to form male reproductive organs. In females, it is the Müllerian ducts that develop into the female reproductive tract while the Wolffian ducts degenerate[6]. In either case, sex determination is controlled by gene expression from the X and Y chromosomes[17,18].

Development of the uterus at week 8–9 commences with the fusion of the Müllerian ducts, which will undergo morphogenesis to form the uterus, fallopian tubes, cervix, and upper vaginal tract. Müllerian duct fusion in humans results in one central uterine cavity, in contrast to rodents, where Müllerian duct fusion is less extensive to allow the formation of two separated uterine horns[11,16]. At week 16, the developing human uterus starts to form glands in the endometrium. Endometrial glands gradually increase complexity and develop branches within the stroma until birth and will continue to develop postnatally until puberty[11]. This contrasts with mouse development where glands are formed exclusively after birth[19,20]. Nevertheless, both human and mouse endometrial gland formation requires WNT signalling[21,22]. Prenatal uterine development concludes with the formation of myometrium at week 22 and the uterus assumes its adult shape[23]. The final steps of human uterus development occur during puberty when the uterus further matures under the influence of sex steroid stimulation and initiates the menstrual cycle.

## Anatomy and function of the uterus

The uterus is located within the pelvic area, between the bladder and the rectum. In humans, it is on average about 7 cm long and weighs around 60 g in the non-pregnant state (non-gravid), extending to up to five-fold in size during pregnancy (gravid). The uterine body contains a triangular uterine cavity with the isthmus leading to the cervix, which connects to the vaginal opening (Fig. 2a). The uterus is supported by uterine ligaments which, together with the ovaries and fallopian tubes, form the appendages of the uterus. Externally, the uterus consists of thick, smooth muscle bundles, the myometrium, and is covered by serous tissue termed perimetrium (Fig. 2b).

The endometrium is the inner mucosal layer, which surrounds the uterine cavity and is comprised of stroma and uterine glands. The endometrium changes cyclically in terms of function and appearance during the menstrual cycle (Fig. 2c). Apes, Old World monkeys, and some New World monkeys undergo a menstrual cycle characterised by external bleeding due to shedding of the outermost layer (menses)[24]. Most other mammalian species experience an oestrous cycle, in which the uterus undergoes remodelling throughout the cycle without shedding[25]. During human menstruation, only the endometrial *functionalis* layer facing the uterine cavity is shed while the *basalis* layer situated towards the myometrium remains unaffected by hormonal changes (Fig. 2b)[26–28]. The human menstrual cycle lasts on average 28 days and is orchestrated by steroidal hormones, including oestrogen and progesterone (Fig. 2c). Following menses, oestrogen is secreted by the ovaries, stimulating endometrial proliferation, which rebuilds the shed surface layer in the proliferative, or follicular, phase. The egg is released around day 14 of the cycle during ovulation. Progesterone secreted in the ovaries from the remnants of the follicle which contained the released egg, the corpus luteum, stimulates the thickening of the uterine lining and initiates the secretory, or luteal, phase. The

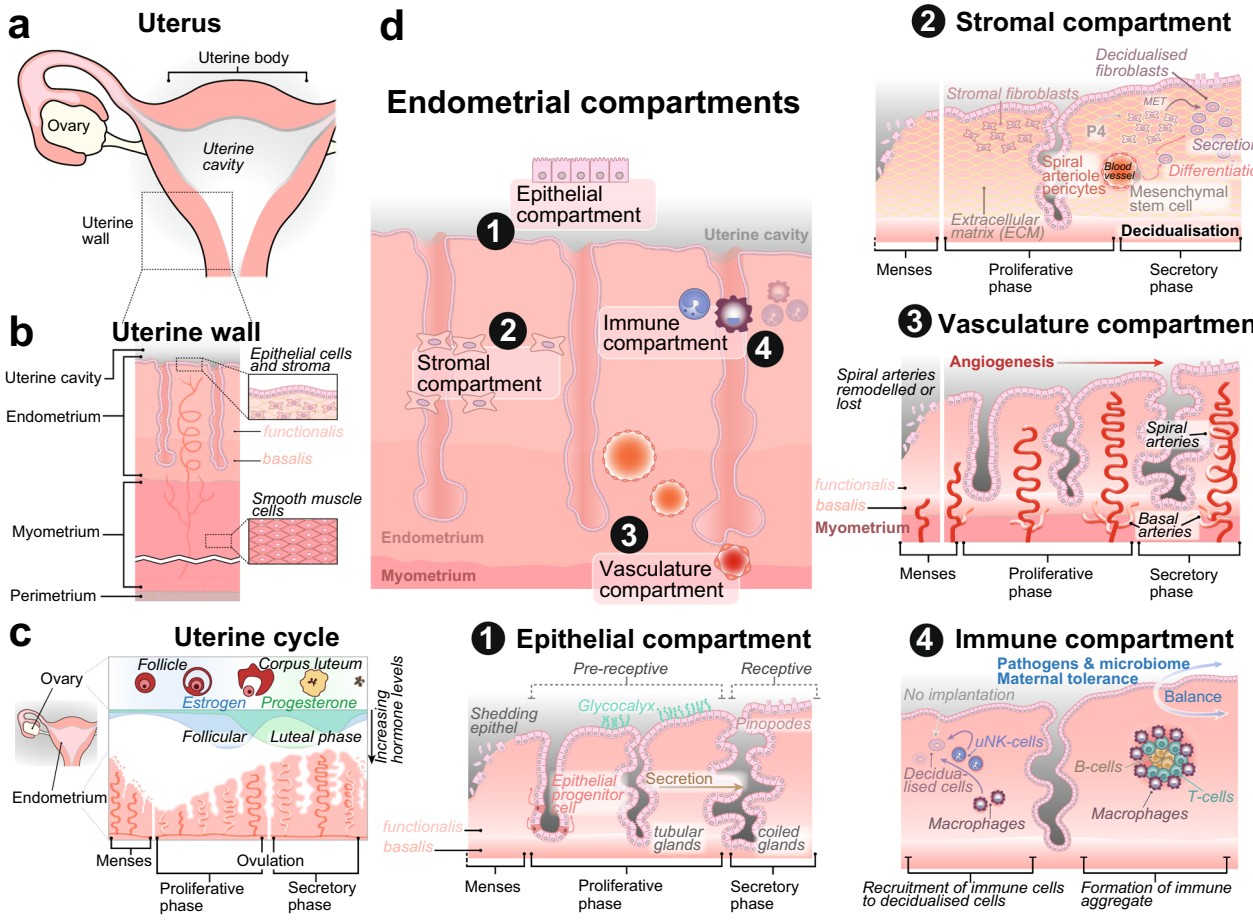

**Fig. 2 Anatomical structure of the uterus. a** The cavity of the human uterine body connects to the ovaries by the fallopian tubes. **b** The uterine wall consists of the endometrium, the smooth muscular myometrial layer, and the outermost perimetrium. The endometrial epithelium forms the border to the cavity (luminal epithelium) and deep-reaching endometrial glands (glandular epithelium) which are embedded in the stroma. Only the endometrial functionalis layer is shed during menstruation at the beginning of the human uterine cycle (**c**), while the non-hormone-responsive basalis layer remains intact. Oestrogen produced by the developing follicle in the ovaries induces the growth and proliferation of endometrial tissue during the proliferative (follicular) phase. Ovulation and release of the egg from the ovaries occur mid-cycle, which leads to increased progesterone levels during the secretory (luteal) phase. Degradation of the corpus luteum and decreasing hormone levels induce menses and re-start the uterine cycle if no embryo implants into the uterine wall. **d** The endometrium can be functionally distinguished into four compartments, which change throughout the uterine cycle. The luminal epithelium of the epithelial compartment (1) is shed during menses and rebuilt by differentiating epithelial progenitor cells migrating from uterine glands. The luminal epithelium becomes pre-receptive but avoids pre-mature attachment of an embryo by the formation of a highly charged glycocalyx layer. Uterine glands grow in size and increase secretion towards the secretory phase while changing shape to become coiled. The luminal epithelium grows apical cellular protrusions (pinopodes), which increases receptivity and allows embryo attachment and implantation. Decidualisation occurs during the secretory phase in the stromal compartment (2) and assists in the preparation of the endometrium for implantation. Mesenchymal stem cells surrounding spiral arteriole (specialised uterine blood vessels) pericytes differentiate to become secretory. Concomitantly, stromal fibroblasts undergo mesenchymal-to-epithelial transition (MET) under the influence of progesterone (P4) and convert to decidualised, secretory fibroblasts. Spiral arteries in the vascular compartment (3) branch from basal arteries, which supply the endometrial basalis layer during menses, when arteries of the functionalis layer are extensively remodelled or lost. Angiogenesis drives the formation of spiral arteries towards the secretory phase to increase the blood supply of the upper endometrium in preparation for embryo implantation. The uterine immune compartment (4) is a highly flexible system. Immune aggregates consisting of B-cells, T-cells and macrophages form and reside in the upper parts of the functional layer, maintaining the balance between defence against pathogens and external threats and sustaining the protective microbiome barrier necessary for uterine health. During pregnancy, the immune compartment provides maternal tolerance to an embryo which induces attachment to the uterine wall. If no successful implantation occurs, immune cells such as uterine natural killer (uNK) cells and macrophages are recruited to remove decidualised cells.

progesterone signals from the corpus luteum induce stromal cells to undergo a transformation, termed decidualisation, in preparation for embryo implantation. In case of successful fertilisation, foetal trophoblast and maternal corpus luteum cooperatively sustain hormone levels to maintain the pregnancy, otherwise, with progesterone levels declining, the outermost endometrial *functionalis* layer is lost as a result of menstruation[26–28].

The *functionalis* and *basalis* layers of the endometrium are composed of four tissue compartments (Fig. 2d):

1. *Epithelial compartment*: The luminal epithelium establishes the boundary to the uterine cavity and is remodelled during the menstrual cycle to become receptive for the implanting embryo. Uterine glands consist of connected glandular epithelium, which extends towards the myometrial border and cyclically change shape from tubular, in the proliferative phase, to coiled, in the secretory phase. The luminal epithelium is lost during menstruation, and the stroma is exposed to the uterine cavity. This leads to the

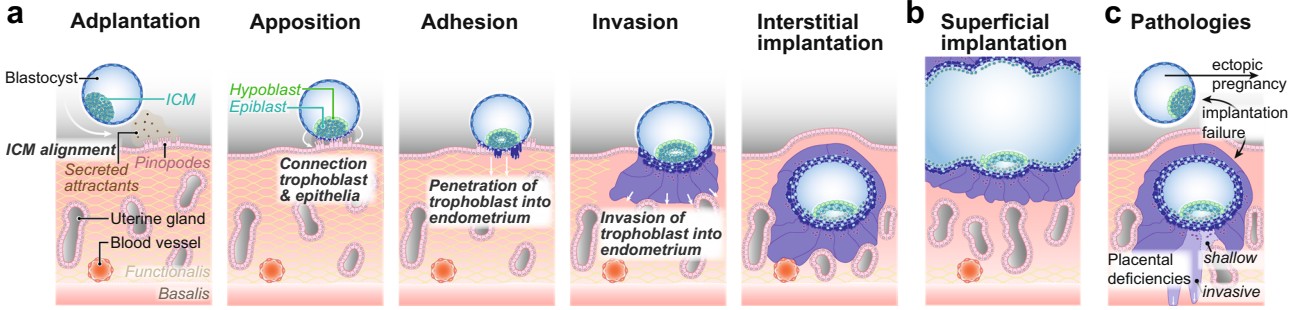

**Fig. 3 Embryo development within the uterus and uterine pathologies. a** The stages leading to human interstitial embryo implantation and marmoset superficial implantation (**b**) are shown from left to right. The blastocyst arrives in the uterine cavity and hatches from the surrounding zone pellucida by day 5 post-fertilisation. It then moves along the uterine epithelium until it reaches a receptive environment provided by pinopodes before the inner cell mass (ICM) aligns towards the uterine epithelium (adplantation). The outer extraembryonal trophoblast cells of the blastocyst connect to the uterine cells (apposition) and penetrate through the luminal epithelium (adhesion). Concomitantly, the ICM differentiates into extraembryonal hypoblast and embryonal epiblast, which induce lumenogenesis. Trophoblast comprises of several layers of proliferative cytotrophblast (dark blue) and terminally differentiated, multinucleated syncytiotrophoblast (purple) which continue invasion into the endometrium towards nutrition-providing uterine glands and blood vessels, by degrading the extracellular matrix of the decidualised stroma. The human embryo is fully embedded in maternal endometrium following implantation. In contrast, the embryo is not fully engulfed by endometrium in superficial implantation (**b**) but stays within the uterine cavity. Common pathologies (**c**) associated with early stages of pregnancy are the failure of an embryo to implant into the uterine cavity wall, or ectopic pregnancy, when the embryo does not reach the uterine cavity for implantation, and instead implants mostly within the uterine tube. **d** Abnormal invasiveness causes placental deficiencies leading to placenta accreta spectrum disorders, caused by excessive trophoblast invasion, or pre-eclampsia, as a result of insufficient trophoblast invasion.

transformation of glandular epithelial cells, which migrate towards the cavity and rebuild the luminal epithelium[26].

2. *Stromal compartment*: The endometrial stroma is a connective tissue consisting of fibroblasts and extracellular matrix. In the mid-secretory phase, fibroblast-like stromal cells differentiate into rounded epithelioid-like cells in response to progesterone, a process termed decidualisation. Human decidualisation begins when stromal cells surrounding spiral arterioles in the upper two-thirds of the endometrium enlarge 6 days after ovulation in preparation for a potential pregnancy.

3. *Vasculature compartment*: Basal arteries branching from the uterine radial arteries supply the *basalis* layer and produce a capillary bed. This supports the *functionalis* layer viability during menstruation[29], when its spiral arteries are extensively remodelled and partially lost. After menstruation, spiral arteries are rebuilt by angiogenesis. Progesterone levels control the blood flow, which increases in the secretory phase, and decreases towards the onset of menstruation. In the absence of progesterone, the blood supply to the spiral arteries stops, causing the stroma to become necrotic and ultimately leads to the onset of menstruation.

4. *Immune compartment*: Accounting for an estimated 10–15% of the stromal compartment, the endometrial immune compartment provides protection against pathogens and balances the commensal microbiome[30,31]. Importantly, it adopts an immunosuppressive state during embryo implantation and acquires maternal tolerance in reaction to hormonal changes[32]. The composition of the immune compartment changes significantly during the menstrual cycle[33], including a substantial increase in uterine natural killer cells following ovulation, which represent the dominant uterine leucocytes during pregnancy.

Pregnancy is established when a competent blastocyst implants into the receptive endometrium. Embryo implantation begins with adplantation and apposition of the blastocyst at the site of implantation, allowing trophoblast cells to attach (Fig. 3a).

Multinucleated syncytiotrophoblast penetrates the luminal epithelium of the endometrium, crosses the basement membrane, and invades the stromal compartment. In human and Great Apes, the conceptus burrows into the endometrium, resulting in an interstitial implantation, while in Old World and New World monkeys, the embryo remains within the uterine cavity and undergoes superficial implantation[34,35] (Fig. 3b). After the initial stages of embryo implantation, human cytotrophoblast and differentiated extravillous trophoblast further invade and induce remodelling of the spiral arteries and maternal tissues. Decidualisation plays an important role in mediating the invasiveness of the trophoblast cells and subsequent placentation via the secretion of chemoattractants. Whilst in humans, decidualisation occurs naturally during the menstrual cycle and develops further following implantation, in Old World and New World monkeys decidualisation is initiated post-implantation and may be less extensive, or proceed more gradually[34,35]. In Old World monkeys, following initial attachment, but prior to the invasion, proliferation and hypertrophy of the epithelium occur at the periphery of the invading syncytiotrophoblast. The endometrial epithelium adjacent to the implantation site remodels its glandular appearance and undergoes a transformation into epithelial plaques[35,36]. Epithelial plaque formation does not occur in humans or Great Apes but has been observed in New World monkeys, including the marmoset[34,37]. The mechanisms controlling the different degrees of trophoblast invasion in monkeys and apes remain an intriguing open question and are highly relevant for pathologies affecting placental invasion.

## Pathologies of the gravid and non-gravid uterus
Human embryo implantation is an inherently inefficient process. Only around 25% of blastocysts implant into the uterus, representing a major hurdle to natural conception[38–40] and rate-limiting step in assisted reproductive technologies[41,42]. In 2% of pregnancies, the embryo implants outside of the endometrium, often into the fallopian tubes, resulting in an ectopic pregnancy[43]. This can be life-threatening for the mother if left untreated and requires termination of the pregnancy. In about 2–8% of pregnancies, the trophoblast fails to invade the uterus properly,

leading to pre-eclampsia[44,45], which is characterised by high maternal blood pressure. Pre-eclampsia can result in seizures and, consequently is one of the leading causes of maternal morbidity and mortality worldwide with severe risks for both mother and the unborn child[46]. In 1 in 533 births[47], excessive trophoblast invasion into deep maternal uterine tissues impairs separation of the placenta from the uterine wall following delivery[48] (Fig. 3c). These *placenta accreta spectrum* disorders cause major obstetric maternal haemorrhage and sometimes require an emergency hysterectomy (removal of the uterus. *Placenta accreta spectrum* disorders remain a major cause of maternal death, with a mortality rate of up to 7%[49]. Antenatal diagnosis is limited to ultrasound methods and MRI, resulting in up to two-thirds of cases remaining undiagnosed until delivery. These pathologies are responsible for only a fraction of all pregnancy loss, and it is estimated that up to 1 in 4 pregnancies will result in a miscarriage[40–42]. Advanced in vitro models of the uterus should provide a strong basis for investigating the mechanisms of maternal–embryo interactions, and subsequent pregnancy loss.

Pathologies of the non-pregnant uterus include conditions relating to menstruation (dysmenorrhoea, menorrhagia), musculature (uterine prolapse), and genetic malformations (uterine septum, bicornuate uterus). There are around 9400 cases of endometrial cancers each year in the UK, with a 92% 5-year survival rate when diagnosed at the earliest stage compared to 15% when diagnosed at stage four[50]. The most common treatment for endometrial cancers is hysterectomy, potentially combined with radiotherapy or chemotherapy. Hysterectomy may also be necessary for other forms of cancers, including ovarian and cervical cancers, and gynaecological sarcoma. Endometriosis is a chronic condition affecting 10% of the female population[51]. It occurs when cells of the endometrium grow outside of the uterus[52], most commonly in the ovaries and fallopian tubes. These cells remain hormone-responsive during the menstrual cycle, resulting in inflammation of the surrounding regions and eventual formation of scar tissues. Although there are strong heritability[53,54], the risk factors and mechanisms remain unclear. There is currently no cure and treatment is limited to surgery, hormone treatment, and pain relief. Stem cell-based disease models have the potential to increase our mechanistic understanding of these pathologies, and thus advance clinical and regenerative treatments.

## Building blocks for a stem cell-based uterus

The assembly of a stem cell-based uterus requires developmentally authentic cell lines as constituent parts (Fig. 4). Previous in vitro models of endometrium rely on cancer-derived cell lines[55] which are likely to be compromised due to their carcinogenic origin. Consequently, we suggest that the construction of stem cell-based uteri should be based on either pluripotent stem cells (PSCs) or patient-specific primary cultures, depending on the research objective.

The differentiation of PSCs into the endometrial epithelium and stromal-like cells presents a promising avenue (Fig. 4). The first differentiation attempts were based on co-culture of PSCs with primary endometrial cultures[56,57]. However, these conditions were poorly defined, and the resulting cells were not fully characterised. Miyazaki et al. addressed this problem by differentiating human-induced PSCs (iPSCs) into endometrial stromal fibroblast-like cells in a 14-day embryoid-body-based approach in defined conditions[58]. The protocol closely follows in vivo endometrium development: iPSCs progress from primitive streak through intermediate mesoderm, coelomic epithelium and Müllerian duct-like stages. The first part of the differentiation is based on a protocol used to generate kidney progenitor cells[59].

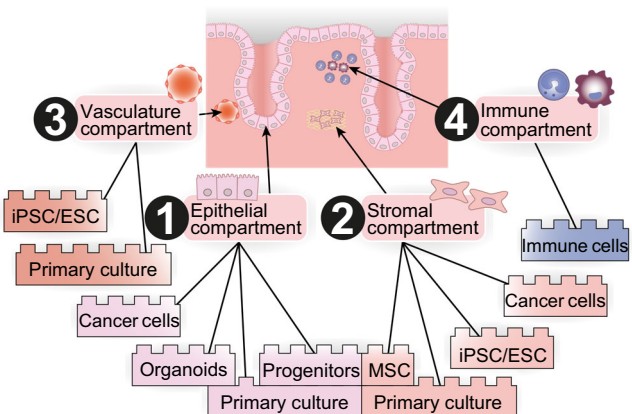

**Fig. 4 The cellular building blocks of a stem-cell-based uterus.** Appropriate cellular models of the epithelial compartment of the uterus (1) can be established as organoids and progenitor cell cultures, which can be derived from primary patient cells. The stromal compartment (2) can be founded by induced pluripotent stem cells (iPSC) or embryonal stem cells (ESC) cultures, or mesenchymal stem cells (MSC) derived from primary cell culture. The vasculature compartment (3) can be derived from iPSCs, ESCs or directly from primary cells, and immune cells of the endometrium build the immune compartment (4).

Intermediate mesoderm-like cells express markers including LHX1 and PAX2. The fact that the intermediate mesoderm-like cells derived by this protocol can be used to make both kidney and endometrial-like cells, lends credibility to the developmental authenticity of the system. BMP inhibition prevents the acquisition of renal identity, while WNT activation is required for Müllerian duct and uterine gland development in vivo[13,14,60,61]. The resulting endometrial stromal fibroblast-like cells were hormone-responsive and resembled primary endometrial stromal cells as determined by RNA sequencing (RNA-seq). This protocol provides an encouraging starting point for iPSC-derived models of human endometrium.

Primary cultures of human adult tissue represent an alternative approach for building a stem cell-based uterus. Endometrium biopsy-derived epithelial and stromal cultures can be maintained in vitro and respond to hormonal stimuli[59,62,63]. Purified endometrial cell cultures have been used to understand physiological and pathological processes, including trophoblast invasion[64]. However, endometrium-derived primary culture offers limited potential for use in in vitro models since they cannot be propagated in vitro for many passages and tend to differentiate[62,63]. Recent advances in organoid culture demonstrate that when epithelial cells from endometrial primary cultures are dissociated, separated from the stromal compartment, and seeded into a 3D-extracellular matrix scaffold, they are capable self-organisation into hollow spheres formed by a single layer of columnar epithelium[65–69]. These spheres are called endometrial organoids as they recapitulate several physiological characteristics of the endometrium, and RNA-seq shows organoids share many similarities with in vivo endometrial glands[66,70]. In contrast to two-dimensional primary cultures, endometrial organoids can be readily propagated, stay genetically and phenotypically stable, and retain the capacity to respond to hormonal stimuli[65,66,70,71]. Strikingly, when exposed to sex hormones, such as progesterone or oestrogen, organoids develop characteristics of early pregnancy and are able to recapitulate the menstrual cycle[65,66,70]. Single-cell profiling identified several sub-populations in endometrial organoids, including proliferating, secretory, ciliated, and putative stem cell populations[70]. A subset of cells harbours the potential to

generate either secretory or ciliated epithelial cells, which is consistent with the proposition of a progenitor population in vivo[66]. Overall, endometrial organoids represent the first step towards 3D-modelling of the endometrium and offer a sustainable source of endometrial epithelial cells[71–74].

## Assembly approaches

Due to the broad range of topics relating to uterine pathologies and maternal–embryo interactions, a single endometrial culture system is unlikely to address all biological questions of interest. Instead, a modular approach is required, with stem cell-based uteri variously assembled from the minimal components necessary to answer key questions[74,75]. Such modularity is advantageous due to its flexibility, whilst the use of minimal component sets avoids unnecessary confounding factors that may obscure interpretation and confers a degree of scalability. Here, we focus on three key approaches: (i) self-organising cultures to model early implantation; (ii) controlled assembly approaches to model trophoblast invasion, placentation, and embryo maintenance, or pathologies of the uterus; (iii) scaffold-based approaches that may be used to model pathologies of the uterus, with potential clinical implications.

## Self-organising co-cultures

The co-culture of embryos alongside endometrial epithelial monolayers has long been suggested as a model for implantation and to gauge mechanisms of apposition and competence of blastocysts for maternal–embryo interactions[76–84]. Similarly, the culture of embryos with endometrial stromal cells provides opportunities for investigating aspects of trophoblast invasion at the earliest stages[82,85–89]. Multi-layered co-cultures combining epithelial and stromal cells offer a more complete model of early implantation[90–92] but are limited by the availability of uterine tissue and embryo samples. Whilst issues with the availability of maternal tissues may be mitigated by appropriate community resources, such as biobanks, embryos are likely to remain the key bottleneck to these applications.

Endometrial organoids[65,66,70,93] represent a suitable advancement over monolayer cultures, allowing for more complete modelling of endometrial tissues including the characteristic glandular structure. Since changes in endometrial vasculature represent a phenotypic readout of implantation, the incorporation of vasculature within organoid models would provide an important improvement (Fig. 5a). Blood vasculature has previously been induced in co-cultures of PSC-derived endothelial cells with pericytes[94,95] or the introduction of mesodermal progenitor cells into organoid cultures[96,97] (reviewed in ref. [98]). It remains to be seen if such protocols can be adapted to allow survival and self-organisation of vascularised endometrial organoids.

Human blastocyst or first-trimester placenta-derived trophoblast stem cells provide an exciting avenue to model the extraembryonic compartment[99–102]. Trophoblast stem cells are capable of self-renewal and differentiation into syncytiotrophoblast and trophoblastic vesicles[103,104]. The latter may provide suitable surrogate models for human blastocysts to overcome the ethical and practical limitations associated with human embryo research. Indeed, the culture of trophoblast spheroids alongside endometrial epithelial cells[105–116] or stromal cells[89,108,117–119] has already been demonstrated. A key step forward would therefore be the co-culture of vascularised endometrial organoids with trophoblast organoids (Fig. 5a). Hurdles to overcome include identification of suitable media that allow for survival and development of these divergent organoid systems, and issues arising from the reversal of the apicobasal polarity, with microvilli

facing towards the organoid lumen, rather than outwards where contact with the embryo-surrogate would occur (Fig. 5a).

PSC-based approaches have also emerged as useful platforms for modelling embryonic and extraembryonic tissues, and may soon represent useful proxies for embryos. In mouse, co-culture of embryonic and extraembryonic cell types allows the formation of embryo-like structures reminiscent of pre-implantation blastocysts[120,121] and post-implantation mouse embryos[122,123] (reviewed in ref. [124]). A combination of embryonic and extraembryonic tissues is likely to be required to faithfully recapitulate maternal–embryo interactions, rather than extraembryonic trophoblast tissues alone (Fig. 5a). Indeed, secreted ligands from the mouse inner cell mass (ICM) induce proliferation in adjacent polar trophectoderm[125,126] and play essential roles in implantation. This might be of particular relevance for primate development, where the blastocyst implants with the ICM oriented towards the endometrium[127,128]. Continuing advances in the field will provide stem cell-based embryo models, which are bound to outperform models based on trophoblast organoids alone. Ultimately, the establishment of robust stem cell-based embryos in human and non-human primates will be an essential goal for the development of realistic and scalable models of early pregnancy.

Selective incorporation of components of the immune system[129–131] represents an additional advantage of assembled stem cell-based uteri. The importance of immune cells in preventing diseases[33,132] as well as their immunomodulatory role during pregnancy[132–139] is well established, and the distribution of immune cells is highly dynamic within the menstrual cycles (reviewed in ref. [33]). The inclusion of immune components into organoid models would be an asset to delineate mechanisms of immune tolerance, albeit at the expense of increased complexity. Careful consideration should be given as to which components of the immune system to include in order to retain tractability of the organoid approaches. The abundance of uterine natural killer cells within the uterus during pregnancy may present a natural starting point. Due to the plasticity of components of the immune system care must be taken to minimise undesired transdifferentiation of immune components in culture. Fluorescence labelling and live-cell imaging will allow tracking of immune cell movements and, combined with single-cell profiling endpoint analysis, will provide candidate regulators for immunomodulatory functions of the uterus.

## Controlled structural assembly: layered deposition and organ-on-a-chip

Decidualisation and placentation represent protracted biological processes that extend over large spatial scales. Trophoblast invasion, which is an important part of placentation, has been quantified using invasion assays. Traditional approaches include the culture of trophoblast cells in Matrigel-coated trans-well inserts with their invasiveness being monitored. These techniques are limited in scope and incapable of capturing the complex interplay between trophoblast and maternal tissues. Full modelling of trophoblast invasion, placentation, and pathologies such as placenta accreta, requires tissue constructs more akin to the maternal interface.

Print-based technologies have been suggested for a variety of tissue constructions[140,141] including muscle[142] and neural tissues[143]. These approaches are based on the layered deposition of appropriate gel-embedded cells or bio-inks by 3D printing technologies[144]. The increased spatial scale of these tissue constructs over organoids may require the integration of functional vasculature for oxygen and nutrient supply. Blood vasculature has already been induced in Matrigel and hyaluronic acid (HA)-based hydrogels[145,146] and the deposition of epithelial-cell-laden

**a** Self-organising co-cultures

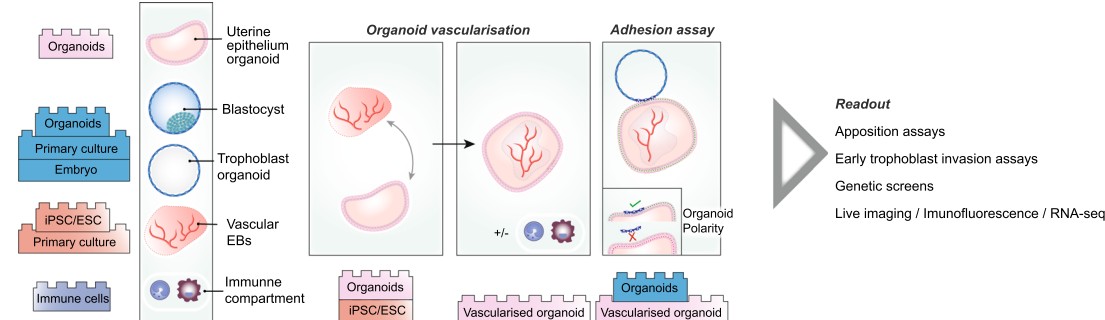

**b** Controlled structural assembly: layered deposition

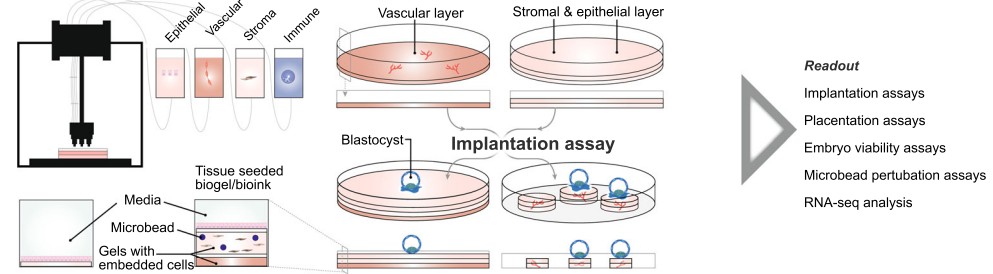

**c** Controlled structural assembly: organ-on-a-chip

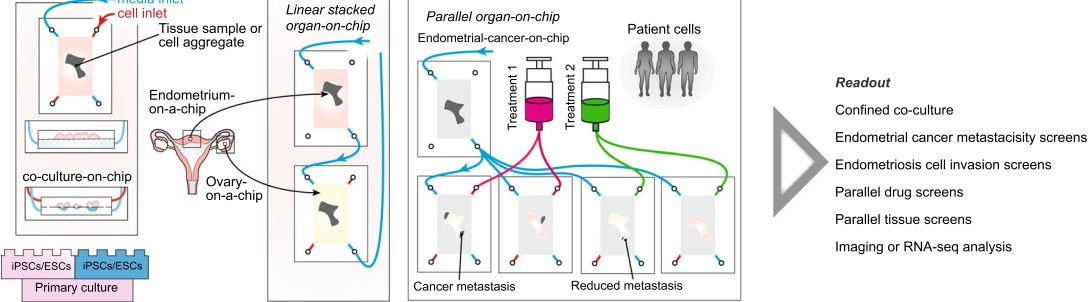

**d** Re-cellularisation of *in vivo* scaffolds

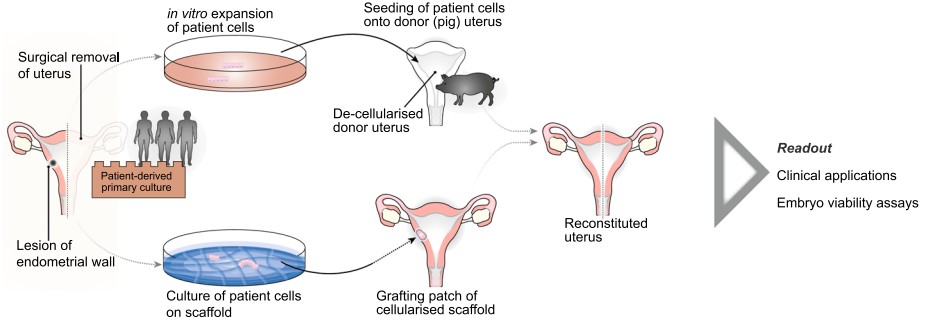

**Fig. 5 Engineering of a stem cell-based uterus. a** Co-culture systems to model the uterus during embryo implantation. Advances include vascularisation of endometrial organoids and integration of immune components. The co-culture of vascularised organoids alongside embryo should allow apposition assays if the polarity of organoids can be inverted. **b** Apposition, adhesion and trophoblast invasion could be modelled in complex tissue constructs engineered by layered deposition of 3D cell-embedded gels using 3D printing. **c** Automatised culture can be obtained by using organ-on-a-chip technology, allowing high-throughput screening of patient-derived uterine cells for co-culture and cell-invasion assays, but also enable linear stacking of multiple components of the female reproductive tract on-a-chip, such as ovarian and fallopian tube-cell containing chips. **d** Pathologies associated with the human uterus, such as a medically required surgical removal of the uterus or a lesion of the endometrial wall following cancer treatment, could be targeted with stem cell-based uteri. Patient-derived primary cells could be expanded in vitro, and seeded onto a de-cellularised donor uterus in order to avoid immune reaction of the patient following re-grafting of the donor uterus. Patient primary cell culture on a scaffold allows to obtain a cellularised grafting patch which can be transplanted to reconstitute the patient's uterus.

hydrogel channels within printed constructs effectively allows the formation of vasculature[147–150]. Another exciting approach entails sacrificial networks, which are cell matrices with embedded dissolvable filaments. Upon the breakdown of these filaments, the created space can be seeded with endothelial cells to generate defined vasculature[151–155]. In the future, sacrificial networks may be routinely combined with print-based approaches[155,156] (reviewed in refs. [98,157]). A limitation of explicit engineering of vasculature is the decreased complexity of branching compared to natural systems. Alternative approaches include seeding of vascularised cell aggregates[96,97] onto a basal scaffolding on which subsequent layers of stromal and epithelial cell-laden gels or organoid-containing gels are deposited (Fig. 5b). To increase parallelisation for screening purposes, constructs could feasibly be engineered over micro-patterned plates[158,159], or alternatively by the embedding of multiple vascularised endometrial organoids within gels that then provide the appropriate signals for invasion (Fig. 5b). A key advantage of print-based approaches is the flexible inclusion of localised signalling by incorporation of chemically laden microbeads within gels, e.g., angiogenic signals to promote vasculogenesis or angiogenesis, or candidate signals to modulate trophoblast invasion.

Functional models of the uterus in different primate systems could be applied to interrogate species-specific differences in the mode of invasion, specifically allowing comparative studies of superficial versus interstitial implantation. Optimisation of such models to allow long-term survival of stem cell-based embryos will provide a powerful asset for investigating the causes of first trimester miscarriages.

Microfluidic "organ-on-a-chip" concepts[157,160–162] in which cell-aggregates or primary sample tissues from a specific organ are cultured within a microfluidic device represent an alternative platform for modelling of the uterus[162] (Fig. 5c). In particular, endometrium-on-a-chip has been used to combine primary human stromal cells with epithelial cells to allow hormone-responsive differentiation of stromal cells into decidua[163]. An important advantage of custom microfluidic devices is the ability to co-culture different cells or cell aggregates at an interface whilst exerting precise control over the signalling environment. This is exemplified in the recent generation of embryonic disc-like and amnion-like cells in structures reminiscent of the early human post-implantation embryo[164]. Modifications of these microfluidic devices would allow for co-culture of layers of endometrial stromal and endometrial epithelial cells, alongside trophoblast cells, each with cell-specific signalling environments. This approach could be used to monitor trophoblast invasion into endometrium on-a-chip (Fig. 5c).

Modularity is one of the key strengths of microfluidic devices. This was recently shown in a model of the female reproductive tract by linear integration of chips containing tissues from fallopian tubes, the uterus, and cervix with shared perfusion between modules[165]. This system supported murine ovarian follicles to produce the human 28-day menstrual cycle and demonstrated the feasibility of modelling complex multi-cellular behaviour by integrating a system of simpler component parts. Extensions of this experimental setup may be of particular use in disease modelling e.g., to test hypotheses about the mechanisms of endometriosis[52]. The linear combination of a chip with endometrial tissues alongside chips containing other tissues with a shared circulating perfusion should provide a powerful assay to gauge retrograde menstruation under hormone varying environments using cells from patients with and without endometriosis. Complex parallel arrays may provide further flexibility, for example allowing a culture of endometrial cancer samples with downstream samples of disease-free endometrium and adjacent organ tissues and one-way connective perfusion, which could be used to investigate metastasis of cancer. Indeed, a single metastatic tumour sample could be connected to multiple identical downstream tissue samples, each with different drug treatment, thus allowing parallelised tissue-specific drug screening (Fig. 5c).

## Re-cellularisation of in vivo-derived tissue scaffolds

A variety of uterine conditions require surgical interventions, including complications during pregnancy. In some cases, previous surgical procedures, including caesarean sections, may impact the ability of the uterus to support a foetus to full term or serve as an aggravating factor in other uterine conditions[47–49]. Stem cell-based models may provide a starting basis for regenerative approaches to address these issues.

Partial reconstruction of the uterus has been achieved using tissue grafts[166] or xenogeneic tissue grafts[167]. A recent study in rabbits restored the function of rabbit uteri following extensive excision of the endometrium using a biodegradable polymer scaffold seeded with autologous primary cells[168] (Fig. 5d). This approach yielded increased implantation rates and sustained pregnancy to term, in contrast to unseeded scaffolds or no-scaffold controls. Whilst graft-based approaches may prove useful for the repair of the uterus following trauma, they cannot address severe pathologies where a hysterectomy may be required. Uterus transplantations are one possibility[169–172] and have allowed successful livebirths[173–175]. However, transplantations are limited by the availability of organ donors. Alternatively, recellularisation approaches in which appropriate cell lineages are seeded onto decellularised scaffolds, have been developed in animal models for lungs[176–180] (reviewed in ref. [181]), heart[182–184], liver[185], oesophagus[186], trachea[187], bladder[172,188], liver[189], and vagina[190]. Decellularisation of rat and pig uteri has been shown to preserve vasculature[191,192], and allow for preliminary recellularisation using neonatal rat uterine cells and rat mesenchymal stem cells[192] or mixtures of stromal and epithelial stem cells from human endometrial origin[191]. Studies in sheep have systematically compared different decellularisation approaches of uteri for their ability to support recellularisation of sheep PSCs[189]. The possibility of using animal scaffolds presents an opportunity for clinical settings (Fig. 5d). However, notable issues associated with the removal of xenoantigens and risks of transmission of animal viruses remain[193]. Finally, the ability of donor uteri from deceased patients to sustain live births[174] may open up the possibility of donor scaffolds. Together, these early studies suggest that repair or reconstruction of uteri using pathogen-free, patient-specific cells may represent a long-term treatment solution to some of the most severe uterine pathologies.

## Outlook

Investigation of the earliest stages of implantation and maternal–embryo interactions with human embryos has long been limited by the 14-day rule[194]. Stem cell-based approaches to mimic the maternal environment synergise with recent advances in modelling embryogenesis[124], and together will illuminate the critical stages of human periimplantation development. Modelling primate implantation with stem cell-based uteri is charged with biomedical potential and provides a unique opportunity for basic science. The development of multi-organoid systems and controlled structural assembly of uterus models should facilitate large-scale genetic and drug screening, enable cross-species comparative studies, and shed new light in the field of human embryogenesis that has previously been obscured. Stem cell-based uterus and embryo research will raise important ethical questions that will require careful consideration by the research community and beyond. Ultimately, advances in basic science go hand-in-hand with progress in clinical applications, and the next few years

promise to see striking steps forward in the advancement of women's reproductive health.

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

## Acknowledgements
We thank the members of the Boroviak lab, in particular Dylan Siriwardena, Erin Slatery, and Max Lycke, as well as Geraldine Jowett and Christos Kyprianou for fruitful discussions and helpful comments on the manuscript. This research is generously supported by the Wellcome Trust (WT RG89228), the Centre for Trophoblast Research, and the Isaac Newton Trust.

## Author contributions
C.A.P., T.E.B., S.B., M.S. and C.M. contributed to conceptualisation and writing of the manuscript. S.B., M.S. and C.M. created figures with input from all authors.

## Competing interests
The authors declare no competing interests.
