## [Peer Review File · Communications Biology]

Reviewers' comments:

Reviewer #1 (Remarks to the Author):

In this review, Bergmann et al. have written a timely review describing various approaches for building a synthetic uterus. Overall, the review is very well written, easy to follow and supported by excellent figures. I have no major suggested changes.

Few minor comments:

1. Ref 51, add Arango et al Dev Biol 2005 as they were first to show the direct role of b-catenin in Mullerian duct development.

2. Please cite Eritja et al J Cell Science 2012, Eritja et al Am J Pathology 2012. They showed 3D growth in the primary endometrial cells before these structures were described as organoids.

Reviewer #2 (Remarks to the Author):

OVERALL COMMENTS

This manuscript provides a very complete and well done, for the most part, review of the current state of building a synthetic uterus. Careful review generated a number of major and specific comments that need to be addressed by revision.

MAJOR COMMENTS

- (1) Review would be better facilitated by sequential line numbering. It is frustrating to review without sequential line numbers or even page numbers.
- (2) There are several spelling and grammar errors that need to be fixed.

SPECIFIC COMMENTS

- (1) The uterine lining is not resorbed in mammals with an estrous cycle. This statement needs to be reworded to indicate it changes thickness, but is not shed as in menstruating species.
- (2) Remove all citations and references to manuscript under review.
- (3) Need to indicate that epithelial cells are seeded as endometrial primary cultures to form organoids (Building blocks for a synthetic uterus section). Stromal fibroblast cells will not do this feat.
- (4) Figure 3a: epithel should be epithelia. This figure is also misleading, as there are many many glands in the endometrium. Indeed, Graham Burton and Berthold Huppertz have microscopic data indicating that embryos will implant adjacent to or within the glands. Please revise the figure.
- (5) References: Please ensure that current topical reviews on this subject are cited, such as:
 - a. Gu ZY, Jia SZ, Liu S, Leng JH. Endometrial Organoids: A New Model for the Research of Endometrial-Related Diseases†. Biol Reprod. 2020 Oct 29;103(5):918-926. doi: 10.1093/biolre/ioaa124. PMID: 32697306; PMCID: PMC7609820.
 - b. Fitzgerald HC, Schust DJ, Spencer TE. In vitro models of the human endometrium: evolution and application for women's health. Biol Reprod. 2020 Oct 3;ioaa183. doi: 10.1093/biolre/ioaa183. Epub ahead of print. PMID: 33009568.
 - c. Ojosnegros S, Seriola A, Godeau AL, Veiga A. Embryo implantation in the laboratory: an update on current techniques. Hum Reprod Update. 2021 Jan 7:dmaa054. doi: 10.1093/humupd/dmaa054. Epub ahead of print. PMID: 33410481.
 - d. Cui Y, Zhao H, Wu S, Li X. Human Female Reproductive System Organoids: Applications in Developmental Biology, Disease Modelling, and Drug Discovery. Stem Cell Rev Rep. 2020 Dec;16(6):1173-1184. doi: 10.1007/s12015-020-10039-0. Epub 2020 Sep 14. PMID: 32929605.

Response to Reviewer's comments
on
“Building a primate synthetic uterus”

Reviewer #1 (Remarks to the Author):

In this review, Bergmann *et al.* have written a timely review describing various approaches for building a synthetic uterus. Overall, the review is very well written, easy to follow and supported by excellent figures. I have no major suggested changes.

Few minor comments:

- a. Ref 51, add Arango *et al* Dev Biol 2005 as they were first to show the direct role of b-catenin in Mullerian duct development.
- b. Please cite Eritja *et al* J Cell Science 2012, Eritja *et al* Am J Pathology 2012. They showed 3D growth in the primary endometrial cells before these structures were described as organoids.

We thank the reviewer for their positive review of our manuscript and for the suggested references. We agree the works they brought to our attention were warrant inclusion in the review and have cited them accordingly, along with an additional reference to Eritja *et al.* (2010) which details the protocol outlined in Eritja *et al* (2012).

Eritja, N. *et al.* A novel three-dimensional culture system of polarized epithelial cells to study endometrial carcinogenesis. *Am. J. Pathol.* 176, (2010).

Reviewer #2 (Remarks to the Author):

OVERALL COMMENTS

This manuscript provides a very complete and well done, for the most part, review of the current state of building a synthetic uterus. Careful review generated a number of major and specific comments that need to be addressed by revision.

We would like to thank reviewer #2 for their positive review of the manuscript and believe that we have fully addressed their comments.

MAJOR COMMENTS

(1) Review would be better facilitated by sequential line numbering. It is frustrating to review without sequential line numbers or even page numbers.

We have added sequential line numbering and page numbers in the manuscript to facilitate evaluation of any changes made in response to both reviews.

(2) There are several spelling and grammar errors that need to be fixed.

We have further proofread the manuscript and sought additional input from native speakers to fix a number of spelling and grammar points throughout the manuscripts.

SPECIFIC COMMENTS

(1) The uterine lining is not resorbed in mammals with an estrous cycle. This statement needs to be reworded to indicate it changes thickness, but is not shed as in menstruating species.

The reviewer is correct in pointing out the need to clarify this statement. The endometrium in estrous mammals is remodelled during the cycle rather than reabsorbed and we have updated our manuscript to clarify this point, including recent reviews on the matter. The corresponding text now reads (see line no. 78-82):

Apes, Old World monkeys, and some New World monkeys undergo a menstrual cycle characterised by external bleeding due to shedding of the outermost layer (menses)²⁴. Most other mammalian species experience an estrous cycle, in which the uterus undergoes remodelling throughout the cycle without shedding²⁵.

This includes the following recent reviews in the area:

Catalini, L. & Fedder, J. Characteristics of the endometrium in menstruating species: Lessons learned from the animal kingdom. *Biology of Reproduction* vol. 102 (2020).

Billhaq, D. H., Lee, S. H. & Lee, S. The potential function of endometrial-secreted factors for endometrium remodeling during the estrous cycle. *Animal Science Journal* vol. 91 (2020).

(2) Remove all citations and references to manuscript under review.

We have removed citations to manuscripts under review as suggested.

(3) Need to indicate that epithelial cells are seeded as endometrial primary cultures to form organoids (Building blocks for a synthetic uterus section). Stromal fibroblast cells will not do this feat.

The reviewer is correct in pointing out that stromal fibroblast cells have not been shown to yield endometrial organoids and are actively filtered out in protocols we reference. We have clarified this in the text (see line no. 222-228):

Recent advances in organoid culture demonstrate that when epithelial cells from endometrial primary cultures are dissociated, separated from the stromal compartment, and seeded into a 3D-extracellular matrix scaffold, they are capable self-organization into hollow spheres formed by a single layer of columnar epithelium⁶⁵⁻⁶⁹. These spheres are called endometrial organoids as they recapitulate several physiological characteristics of the endometrium, and RNA-seq shows organoids share many similarities with in vivo endometrial glands^{66,70}.

(4) Figure 3a: epithel should be epithelia.

We have updated the figure accordingly.

This figure is also misleading, as there are many many more glands in the endometrium. Indeed, Graham Burton and Berthold Huppertz have microscopic data indicating that embryos will implant adjacent to or within the glands. Please revise the figure.

We agree with the reviewer that the number of glands is greater than indicated in the figures, which may be misleading. We have updated our schematic to include more glands to address this point.

(5) References: Please ensure that current topical reviews on this subject are cited, such as:

- Gu ZY, Jia SZ, Liu S, Leng JH. Endometrial Organoids: A New Model for the Research of Endometrial-Related Diseases†. *Biol Reprod.* 2020 Oct 29;103(5):918-926. doi: 10.1093/biolre/ioaa124. PMID: 32697306; PMCID: PMC7609820.
- Fitzgerald HC, Schust DJ, Spencer TE. In vitro models of the human endometrium: evolution and application for women's health. *Biol Reprod.* 2020 Oct 3:ioaa183. doi: 10.1093/biolre/ioaa183. Epub ahead of print. PMID: 33009568.
- Ojosnegros S, Seriola A, Godeau AL, Veiga A. Embryo implantation in the laboratory: an update on current techniques. *Hum Reprod Update.* 2021 Jan 7:dmaa054. doi: 10.1093/humupd/dmaa054. Epub ahead of print. PMID: 33410481.
- Cui Y, Zhao H, Wu S, Li X. Human Female Reproductive System Organoids: Applications in Developmental Biology, Disease Modelling, and Drug Discovery. *Stem Cell Rev Rep.* 2020 Dec;16(6):1173-1184. doi: 10.1007/s12015-020-10039-0. Epub 2020 Sep 14. PMID: 32929605.

We have incorporated a number of recently published papers to our references including the manuscripts suggested above. We have additionally included the following recent endometrial single cell compendium that was released subsequent to our preliminary submission:

Garcia-Alonso, L. *et al.* Mapping the temporal and spatial dynamics of the human endometrium in vivo and in vitro. *bioRxiv* (2021).

We have included the following references that were missed in the earlier draft in reference to the limiting steps in reproductive technologies:

Macklon, N. S., Geraedts, J. P. M. & Fauser, B. C. J. M. Conception to ongoing pregnancy: The 'black box' of early pregnancy loss. *Hum. Reprod. Update* 8, (2002).

Santos, M. A., Kuijk, E. W. & Macklon, N. S. The impact of ovarian stimulation for IVF on the developing embryo. *Reproduction* vol. 139 (2010).

Boomsma, C. M. *et al.* Endometrial secretion analysis identifies a cytokine profile predictive of pregnancy in IVF. *Hum. Reprod.* 24, (2009).

And the following references relating to genetic associations in endometriosis:

Stefansson, H. *et al.* Genetic factors contribute to the risk of developing endometriosis. *Hum. Reprod.* 17, (2002).

Gallagher, C. S. *et al.* Genome-wide association and epidemiological analyses reveal common genetic origins between uterine leiomyomata and endometriosis. *Nat. Commun.* 10, (2019).

Finally, we added the following references relating to artificial uteri..

Partridge, E. A. *et al.* An extra-uterine system to physiologically support the extreme premature lamb. *Nat. Commun.* 8, (2017).

...and the following reference relating to the 14-day rule:

Warnock, M. *Report of the committee of inquiry into human fertilisation and embryology.* vol. 9314 (HM Stationery Office, 1984).